# Cholesteatoma Severity Determines the Risk of Recurrent Paediatric Cholesteatoma More Than the Surgical Approach

**DOI:** 10.3390/jcm13030836

**Published:** 2024-02-01

**Authors:** Adrian L. James

**Affiliations:** 1Department of Otolaryngology—Head & Neck Surgery, Hospital for Sick Children, Toronto, ON M5G 1X8, Canada; adr.james@utoronto.ca; 2Division of Otology & Neurotology, Department of Otolaryngology—Head & Neck Surgery, University of Toronto, Toronto, ON M5S 1A8, Canada

**Keywords:** cholesteatoma, paediatric, recurrence, mastoid obliteration, endoscopic ear surgery

## Abstract

Objective: To evaluate factors that influence the rate of cholesteatoma recurrence (growth of new retraction cholesteatoma) in children. Methods: Review of children with primary acquired or congenital cholesteatoma. Severity was classified by extent and EAONO-JOS stage, and surgery by SAMEO-ATO. Primary outcome measure was 5-year recurrence rate using Kaplan–Meier or Cox regression analysis. Results: Median age was 10.7 years for 408 cholesteatomas from which 64 recurred. Median follow up was 4.6 years (0–13.5 years) with 5-year recurrence rate of 16% and 10-year of 29%. Congenital cholesteatoma (n = 51) had 15% 5-year recurrence. Of 216 pars tensa cholesteatomas, 5-year recurrence was similar at 14%, whereas recurrence from 100 pars flaccida cholesteatomas was more common at 23% (log-rank, *p* = 0.001). Sub-division of EAONO-JOS Stage 2 showed more recurrence in those with than without mastoid cholesteatoma (22.1% versus 10%), with more in Stage 3 (31.9%; *p* = 0.0003). Surgery without mastoidectomy, including totally endoscopic ear surgery, had 11% 5-year recurrence. Canal wall-up tympanomastoidectomy (CWU) and canal wall-down/mastoid obliteration both had 23% 5-year recurrence. Multivariate analysis showed increased recurrence for EAONO-JOS Stage 3 (HR 5.1; CI: 1.4–18.5) at risk syndromes (HR 2.88; 1.1–7.5) and age < 7 years (HR 1.9; 1.1–3.3), but not for surgical category or other factors. Conclusion: Young age and more extensive cholesteatoma increase the risk of recurrent cholesteatoma in children. When controlling for these factors, surgical approach does not have a significant effect on this outcome. Other objectives, such as lower post-operative morbidity and better hearing outcome, may prove to be more appropriate parameters for selecting optimal surgical approach in children.

## 1. Introduction

Cholesteatoma is a growth of keratinising squamous epithelium in the middle ear either from a congenital cyst or acquired from the ingrowth of retraction from the tympanic membrane. Following surgical removal, recurrent cholesteatoma may form from the ingrowth of a new squamous retraction. This differs from residual cholesteatoma which is defined as the growth of a remnant left in situ following incomplete surgical excision [1]. Recurrent cholesteatoma requires revision surgery and may occur many years after surgery, necessitating long term follow up to monitor for this risk [2,3].

It is recognised that recurrent cholesteatoma is more common in children than adults, but the reasons why it recurs in some ears and not others is not well understood [4,5,6]. Contradictory hypotheses have been advanced to try and explain why different surgical techniques might be used to prevent recurrence. Endoscopic trans-canal surgery has been proposed to open tympanomastoid ventilation pathways, preserve mastoid mucosa, and normalise mastoid air pressure homeostasis [7]. Canal wall-up tympanomastoidectomy, via the removal of mucosa and bone from the mastoid, has been said to create a larger air-volume reservoir to buffer changes in middle ear pressure [8]. The opposite strategy of mastoid obliteration has been recommended to minimise negative middle ear pressure generation by eliminating gas absorption from the mastoid [9], but the quality of evidence used to compare the effectiveness of these different intervention has typically been poor [10]. Few have used survival analysis which should be mandatory to control for the increased rate of recurrence with time and for cases that have not been followed for long enough to have developed recurrence [11,12,13]. Few studies have controlled for additional factors that might influence surgical outcome such as the type or severity of cholesteatoma.

The objective of this study is to investigate the relative contributions of type of surgery and type and severity of cholesteatoma on recurrence rates in children. Cholesteatoma is classified into type and severity according to the EAONO-JOS classification [1]. Surgery is classified into the three physiologically distinct categories outlined as follows: (a) surgery through the ear canal without mastoidectomy, (b) canal wall-up tympanomastoidectomy, and (c) elimination of the mastoid by canal wall-down mastoidectomy or obliteration.

## 2. Materials and Methods

Approval for this study was granted by the institution’s Research Ethics Board (study numbers 1000033566, 1000067921, and 1000012951).

Data on characteristics of the cholesteatoma and the details of the surgery and outcome at clinical follow up were collected prospectively on a consecutive series of children up to 18 years of age from 2005 to 2020. Ears presenting after previous cholesteatoma surgery elsewhere were excluded as information on the nature of original cholesteatoma and surgery was incomplete. Secondary acquired cholesteatoma (n = 18) and ear canal cholesteatoma consistent with wax keratosis or keratosis obturans (n = 7) were also excluded as having relatively small numbers. All other cases were included.

Cholesteatoma was classified according to the EAONO-JOS classification into type (congenital, or acquired from retraction of pars tensa, pars flaccida, or both; secondary acquired, or uncertain) and stage [1]. Staging was applied prospectively after publication of the classification in 2017 but retrospectively to earlier data as described previously [14]. To compensate for a preponderance of cases in Stage 2, this group was divided according to whether cholesteatoma extended into the mastoid or not, also as proposed previously [14,15]. Type of surgery was categorised into three types according to presumed effect on mastoid physiology. As defined by SAMEO-ATO nomenclature (S = stage of surgery; A = approach; M = mastoidectomy type; E = ear canal reconstruction; O = obliteration) [16], these three types were as follows:(a)Mastoid-sparing surgery (S1, A1/A4, Mx/M2a/b with E2, Ox). Increasingly since 2008, this was predominantly completed endoscopically trans-canal, but previously through a post-auricular incision with microscope. Removal of scutum (and occasionally part of the posterior canal wall) was completed if necessary but mastoid air cells were not opened. The scutum and ear canal were reconstructed with cartilage, or less often bone paté. Most cholesteatoma allocated to this group would have been confined to the tympanic cavity, although cases of endoscopic trans-canal removal of cholesteatoma from the antrum were also included in this group. The opened tympanic isthmus would allow ventilation between tympanum and mastoid. Mastoid function is expected to be unchanged with this approach although many are not fully aerated post-operatively [17].(b)Canal wall-up tympanomastoidectomy (S1, A4, M1a/b, +/−M2a, E2, Ox). Typically, a relatively conservative cortical mastoidectomy was drilled for access to the limit of the cholesteatoma. Posterior tympanotomy was rarely required as endoscopes were used throughout the series for retrotympanic control. This approach was used when cholesteatoma could not be removed through a trans-canal approach. Mastoid volume is enlarged and mucosal surface area is reduced with this technique, although approximately half become filled with scar tissue [17].(c)Mastoid elimination (S1, A4, M1 or M2c, Ex, O1/2). This group includes ears in which the mastoid air space is disconnected from the middle ear air space. It includes canal wall-down modified radical mastoidectomy (M2c) which often received partial or complete obliteration with autogenous bone paté and soft tissue rotation flap to minimise post-operative cavity maintenance. The group also includes obliteration of canal wall-up tympanomastoidectomy using the technique described by Offeciers [18]. Both canal wall-down and canal wall-up obliteration eliminate the mastoid air space and create a small middle ear space, in principle leaving similar spaces into which recurrent disease could grow. With all the techniques in this group, gas absorption across mastoid mucosa and any pressure buffering effect of the mastoid are removed, so they cannot contribute to middle ear pressure homeostasis and retraction disease [9,19,20]. Predisposition to recurrent cholesteatoma is therefore considered equivalent. Ears allocated to this group would generally have been considered to have more severe disease or risk factors for recurrence [21].

Data were recorded on the appearance of the ear after micro-debridement at follow up, which was offered at least annually to all patients until 18 years of age. Referral to adult services was then made and data on subsequent follow up were included when received (sporadically). Recurrent cholesteatoma was defined as an accumulation of keratin debris from squamous epithelium that was in continuity with the surface of the tympanic membrane and could not be removed with micro-debridement. Usually, this was from a new tympanic membrane retraction pocket. Growth of skin into previously obliterated bone and entrapment of skin from new bone growth across a mastoid cavity were included within this definition. An acquired cholesteatoma from retraction after removal of a congenital cholesteatoma was also classified as recurrence. Residual cholesteatoma (growth of incompletely removed remnants of cholesteatoma following previous surgery) was not included for analysis in this study of recurrent disease.

### Analysis

The primary outcome measure was rate of cholesteatoma recurrence at 5 years. Recurrence rate at 10 years was considered a secondary outcome measure, anticipating that too small a proportion of the group would be followed for reliable investigation of this time period, particularly for subgroup analysis. Survival analysis was completed with event being the time of development of recurrent cholesteatoma. Ears were censored at time of last follow up. Kaplan–Meier log-rank analysis was used to compare individual categorical variables such as type of cholesteatoma. Multivariate analysis of significant factors was completed with Cox proportional hazard regression analysis. Analysis was completed with the survfit and coxph functions of R software (version 4.3.1). Kaplan–Meier plots were compiled according to Kmunicate guidelines [22]. Significance was determined with *p* < 0.05. Age was categorised into three groups. Young age was defined as under 7 years of age based on previous cut-point analysis of the dataset showing increased risk of recurrence under age 6.7 years (unpublished data). Teenagers comprised the oldest age group, based on the untested hypothesis that post-pubertal children might have grown out of younger risk factors, and that this group would have limited opportunity to provide 5-year follow up.

## 3. Results

This series includes 408 cases of cholesteatoma in 383 children (25 (6.1%) bilateral cases). The average age at presentation was 10.7 years (mean and median), with a range of 1.8–17.8 years. When categorising the children into age groups, 69 were considered young (<7 years), and 119 were teenagers (13–18 years). More cholesteatoma presented in boys (265, 65.0%), with slightly more on the left side (226, 55.4%). Those with risk factors for cholesteatoma included cleft palate (44, 10.1%), Down syndrome (7, 1.7%), Turner syndrome (5, 1.2%), and Sotos syndrome (4, 1.0%).

### 3.1. Characteristics of Cholesteatoma

#### 3.1.1. Cholesteatoma Type

The type of cholesteatoma was congenital in 51 (12.5%) ears. Primary acquired cholesteatoma arose from pars tensa retraction in 216 (52.9%) ears, pars flaccida in 100 (24.4%) ears, and both in 20 (4.9%) ears. The origin of the cholesteatoma could not be determined in 21 (5.1%) ears, for example, having some characteristics of congenital disease in association with a perforation, or extensive granulation obscuring the origin.

#### 3.1.2. Cholesteatoma Stage

Using the EAONO-JOS staging system, 65 cases were Stage 1 being confined to the subsite of the ear in which they arose (i.e., mesotympanum for pars tensa and congenital, and epitympanum for pars flaccida). There were 291 Stage 2 cases, in which two or more subsites were involved but without other markers of severity such as erosive or suppurative complications. Cholesteatoma extended into the mastoid in half of Stage 2 cases (n = 144, 49%). Stage 3 comprised 48 cases (including extra-cranial suppurative and erosive complications). There were only two cases of Stage 4 disease with suppurative intra-cranial complications.

#### 3.1.3. Cholesteatoma Surgery

Of the three surgical categories, the groups were divided as follows:Group (a) 224 ears were managed with mastoid-sparing surgery, of which 127 were completed with a totally endoscopic approach.Group (b) 148 were managed with canal wall-up tympanomastoidectomy.Group (c) 36 were managed with mastoid elimination surgery, predominantly with canal wall-down surgery. Of these, 17 were obliterated, 9 partially, and 8 with complete mastoid and epitympanic obliteration. Five received the CWU bony obliteration technique.

### 3.2. Cholesteatoma Recurrence (Univariate Survival Analysis)

Recurrent cholesteatoma occurred in 64 ears at a median of 2.9 years after surgery (range 0.5–9.3 years). For ears with no recurrence, median follow up was 4.6 years (range 0–13.5 years). As demonstrated in Figure 1, the recurrence rate at 5 years was 16.4% (95% CI: 12.1–20.6%) with 146 still at risk. Recurrence rate at 10 years was 28.5% (20.1–36.0%) with 22 still at risk.

#### 3.2.1. Recurrence and Age Group

The youngest age group (<7 years age) had the greatest risk of recurrence at 5 years 25.4% (CI: 13.2–35.9%) when compared to older children (log-rank, *p* = 0.02), as can be seen in Figure 2. The older age groups, 7–13 years at 13.9% (CI: 8.7–18.8%) and >13 years at 16.7% (CI: 5.5–26.5%), were not significantly different. The oldest age group was not followed for long enough to provide 10-year recurrence rates. Ears in children with syndromes associated with an increased risk of cholesteatoma noted above had a higher rate or recurrence at 5 years (35.7%, CI 0.0–58.8%, *p* = 0.01), although confidence intervals were broad and follow up was too short in this group to provide 10-year survival data. Other demographic characteristics, including side of cholesteatoma (log-rank, *p* = 0.2), gender (*p* = 0.99), presence of cleft palate (*p* = 0.94), and presence of bilateral cholesteatoma (*p* = 0.21), did not influence recurrence rates.

#### 3.2.2. Recurrence and Type of Cholesteatoma

As shown in Figure 3, the development of retraction cholesteatoma after congenital cholesteatoma removal occurred in 11% (CI: 0–20.9%) at 5 years and 25.6% (CI: 0–44.9%) at 10 years. This was not significantly different from the recurrence after primary acquired pars tensa cholesteatoma (*p* = 0.68) at 13.7% (CI: 8.1–18.9%) and 19.9% (CI: 12.2–26.9%), respectively. However, the outcome from primary acquired pars flaccida cholesteatoma was significantly worse (*p* = 0.001), with a 5-year recurrence rate of 22.6% (CI: 13.3–30.9%) and a 10-year rate of 46% (CI: 24.4–62.1%). Three cases of recurrence occurred from cholesteatoma arising at both pars flaccida and tensa and another three from cholesteatoma of uncertain origin, providing 5-year recurrence rates of 22.5% (CI: 0.0–42.1%) and 13.2% (CI: 0.0–29%), respectively, with the broad confidence intervals associated with their small group size.

#### 3.2.3. Recurrence and Cholesteatoma Severity

Severity of cholesteatoma, as categorised by the EAONO-JOS staging system, showed least recurrence in Stage 1, with only three cases occurring within 3.3 years of surgery, providing 6.3% (CI: 0–13.0%) 5- and 10-year recurrence rates, as shown in Figure 4. Stage 2 recurrence rates were 15.8% (CI: 10.7–20.6%) at 5 years and 30.2% (CI:18.8–40.0%) at 10 years. Stage 3 recurrence rates were 31.8% (CI: 15.3–45.1%) at 5 years and 43.9% (CI:23.5–58.8%) at 10 years. These differences were significant (*p* = 0.0011). Stage 2 with no cholesteatoma in the mastoid had less recurrence than Stage 2 with cholesteatoma in the mastoid: 10% (CI: 4.1–15.5%) versus 22.1% (CI: 13.5–29.8%) at 5 years and 24.3% (CI: 7.8–37.8%) versus 37.4% (19.7–51.6%) at 10 years (*p* = 0.018). Figure 5 demonstrates the ordinal increase in risk of recurrence from Stage 1 through Stage 2 without mastoid extension, to Stage 2 with mastoid extension, and to Stage 3 (*p* = 0.00027). The two cases of Stage 4 cholesteatoma were followed for 0.8 and 9.4 years and neither developed recurrence.

#### 3.2.4. Recurrence and Category of Surgery

Mastoid-sparing surgery (Group a) had a 5-year recurrence rate of 10.8% (CI: 5.8–15.5%) compared to canal wall-up tympanomastoidectomy (Group b) at 23.0% (14.7–30.6%) and 23.2% (CI: 4.5–38.1%) for mastoid elimination surgery (Group c). The recurrence rate at 10 years was (a) 16.7% (CI: 9.7–24.2%), (b) 58.8% (CI: 25.0–53.9%), and (c) 32.8% (CI: 5.5–52.1). Although these rates are different (log-rank, *p* = 0.005), it is important to consider the effect of other variables on this outcome, as the allocation to a surgical group was skewed to cholesteatomas of different severity. The subgroup analysis of Group a showed no significant difference in the 5-year recurrence rate between totally endoscopic surgery and post-auricular surgery (13.8%, CI: 6.7–20.4%, versus 7.0%, CI: 1.0–13.4%; *p* = 0.28). Of note, a larger cholesteatoma could be removed with the endoscopic approach. The subgroup analysis of Group c showed no significant difference in 5-year recurrence between obliteration and no obliteration (23.0%, CI: 0.0–43.0%, versus 22.1%, CI: 0.0–41.6%; *p* = 0.39), although the numbers were too small and confidence intervals were too broad for a reliable comparison.

### 3.3. Cholesteatoma Recurrence (Multivariate Survival Analysis)

The results of Cox proportional hazard analysis are shown in Table 1. When controlling for other factors associated with a greater risk of recurrence in univariate Kapan Meier analysis, it can be seen that the EAONO-JOS Stage 3 cholesteatoma is more than five times more likely to develop recurrent cholesteatoma than Stage 1, although the confidence intervals are broad. Stage 2 disease extending into the mastoid is also at a higher risk, but not significantly. Syndromes associated with cholesteatoma have more than twice the risk of those without such syndromes, but the number of ears in this group was small, making the reliability of this observation less certain. Children younger than 7 years also have an increased risk of recurrence. Recurrence after pars flaccida cholesteatoma does not appear to be higher than the other types of cholesteatoma when controlling for these factors.

This multivariate analysis finds no significant difference in the risk of recurrence whether the mastoid is left in its natural state (Group a), drilled out (Group b,) or de-functioned by removing this air space (Group c): recurrence rate of cholesteatoma is the same regardless of type of surgical management of the mastoid when controlling for other factors that influence the outcome.

## 4. Discussion

The primary finding of this study is that demographic characteristics of the children and the severity of the cholesteatoma appear to influence the risk of recurrent cholesteatoma more significantly than the type of surgery completed. Multivariate survival analysis is required to determine the relative contributions of these different factors to the risk of recurrence. Children presenting at a young age (<7 years) and those with syndromes known to increase the risk of acquired cholesteatoma development, such as Down syndrome, Turner syndrome, and Sotos syndrome [23], are at particular risk. Of interest, even though cleft palate is also known to increase the risk of developing cholesteatoma [24,25], the risk of recurrence does not appear to be elevated in children with a cleft palate. Similarly, even though cholesteatoma is more common in boys, they do not have a higher risk of recurrence than girls. The different origins of cholesteatoma did not significantly influence the recurrence rate in this series after controlling for other factors, although pars flaccida cholesteatoma appeared to be at a higher risk with univariate analysis. Importantly, acquired retraction cholesteatoma can develop after the removal of congenital cholesteatoma. Although not anticipated from the current understanding of congenital cholesteatoma pathogenesis, this finding has been reported previously from different centres [26,27]. From the author’s observation of the development of retraction cholesteatoma after congenital cholesteatoma, it appears that pars flaccida retraction may be caused by mucosal scar tissue across the tympanic isthmus and tensor fold blocking attic ventilation, which is consistent with the selective epitympanic dysventilation hypothesis [7,28]. The risk of recurrence after the removal of a small cholesteatoma (EAONO-JOS Stage 1: confined to one subsite of the ear) was found to be low in this series. However, larger cholesteatoma continued to recur up to 10 years after initial surgery. Too few children were followed longer than that in this series to comment on whether follow ups should be continued into adulthood, but others have reported later recurrence [2,3].

In contrast to the findings of this series, some other series have failed to show any correlation between EAONO-JOS Stage and the risk of residual or recurrent cholesteatoma [29,30,31,32,33]. Potential explanations include the smaller sample sizes, inclusion of adults with lower risk of recurrence, and shorter follow up in those series. The largest study using survival analysis to evaluate the EAONO-JOS staging system found less recurrence with Stage 1 cholesteatoma; this included some of the current dataset [14]. A common limitation visible in most studies of this staging system is the preponderance of cases allocated to Stage 2 and, fortunately, in many health care jurisdictions, the small number of Stage 4 cases [14]. This skewed distribution limits the utility of any predictive function. Sub-division of Stage 2 has been suggested, for example, by ossicular status or the presence/absence of mastoid extension [14,33]. The extension of pars flaccida cholesteatoma into the mastoid has been shown to increase the risk of recurrence [15]. In this series, the division of Stage 2 into those with and without mastoid cholesteatoma split the group into equal-sized halves and demonstrated a significantly higher rate of recurrent cholesteatoma for those that had extended into the mastoid, showing that this modification may provide greater utility to the staging system. The validation of this finding in series from other centres is warranted. Further subdivision of the mastoid into antral or further extension would also be relevant to the selection of surgical approach as antral disease is potentially amenable to trans-canal (Group a) surgery, whereas more extensive disease requires trans-mastoid surgery (Group b or c) [34]. The incorporation of ossicular status also improves a prediction of the recurrence risk and has the added advantage of predicting hearing outcome [33,35,36]. Potentially, other sub-divisions or reclassifications of the different parameters in Stage 3 might further improve the prognostic utility of this staging system.

Arguably, the most surprising finding of this study is that surgical approach did not appear to influence the risk of recurrent cholesteatoma. Other studies, typically on adult cholesteatoma and without multivariate survival analysis, have shown more recurrence after canal wall-up surgery (referred to here as Group b) than canal wall-down surgery (group c) [11]. However, there is surprisingly little evidence of quality to support this contention: a systematic review of 2060 studies yielded only one that distinguished between recurrent and residual disease and used survival analysis [11]. This study showed less recurrence in canal wall-up (group b) than canal wall-down (group c) [37]. Similarly, a metanalysis of 11 studies comparing canal wall-up with mastoid obliteration found no studies using survival analysis, and, with the limited quality of data available, no difference in recurrence [13]. From the current study, the type of surgery appears less relevant to the outcome: it appears that “bad ears do badly and good ears do well” somewhat independently of how the mastoid is treated surgically. With univariate analysis, cholesteatoma removal without drilling the mastoid was associated with less recurrence than canal wall-up tympanomastoidectomy, which in turn had less recurrence than canal wall-down surgery or mastoid obliteration. However, it must be emphasised that, in keeping with standard surgical practice, the allocation to these different approaches was not randomised but skewed toward less invasive surgery for the smaller less severe cholesteatoma. It is only by controlling for differences in the cholesteatoma (and patient) that these different approaches can be fairly compared. With multivariate analysis, no difference in cholesteatoma recurrence was found between these different approaches regarding the surgical management of the mastoid.

There are few other studies that use survival analysis to compare recurrence rate from different surgical techniques in children. One has shown no difference between totally endoscopic surgery and canal wall-up tympanomastoidectomy [38]. Low rates of recurrence have been reported after mastoid obliteration with either canal wall-down or canal wall-up surgery in small uncontrolled series of children [18,39]. Cholesteatoma severity is not described in these series. In contrast to this study, revision surgery for keratin entrapment in retractions was not included within the definition of recurrence in one of these series [18]. The attrition rate was high in the other series, leading to uncertainty regarding the precision of the findings [39]. A larger series reported similar recidivism rates with and without mastoid obliteration using multivariate survival analysis to control for confounding factors but did not report recurrence separately [40]. The proportion of children with recurrence was higher in the obliteration group. Thus, although mastoid obliteration has been touted as a means to minimise recurrence in children, it is clear from the current series that when using obliteration for more severe paediatric cholesteatoma, the prevention of recurrence cannot be guaranteed. Recognising that trans-canal endoscopic surgery has significantly less post-operative morbidity than post-auricular surgery [41], that the use of endoscopes reduces the risk of residual cholesteatoma [42] while maintaining equivalent perforation closure rates [43], and that there is an improvement in hearing from ossiculoplasty [44], the lack of any detectable difference in recurrence rates with this approach suggests that, when possible, paediatric cholesteatoma should be removed using this approach. More extensive disease will continue to need trans-mastoid surgery with or without mastoid obliteration.

The advantages of this study over previous publications on paediatric cholesteatoma include the relatively large consecutive sample size with prospectively acquired data. Importantly, this includes a distinction made at the time of surgery between the identification of recurrent or residual cholesteatoma. Also, survival analysis to control for variable length of follow up in different groups and multivariate analysis to control for confounding variables were used surprisingly infrequently in the existing literature [10,11]. The data from a single surgeon potentially limit the applicability to other surgeons with different case mixes, allocation to treatments, and techniques. Despite the large sample size, the reliability of some subgroup analyses is limited by the small number of events. Even though multivariate analysis was used to control for confounding variables, it is conceivable that the heterogeneity between the three different surgical groups was not adequately controlled. For example, few if any patients receiving mastoid obliteration would have been candidates for trans-canal surgery without mastoidectomy. It is also feasible that the selection of different parameters in the multivariate analysis would change the significant effect of other variables. To obtain more reliable data, randomised controlled trials with an adequate sample size and length of follow up are unlikely to be feasible to compare surgical techniques. Propensity score matching on pooled data from centres which use different techniques on similar patients may provide a more practicable method to obtain useful comparative evidence.

## 5. Conclusions

Following the removal of cholesteatoma in childhood, recurrence can develop many years later, regardless of the congenital or acquired origin of the initial lesion. This risk is higher in young children, and for those with associated syndromes. More extensive or severe cholesteatoma also appears to be at a greater risk of developing recurrence. Although different forms of mastoid surgery might be expected to have different effects on middle ear pressure homeostasis and consequently influence the risk of subsequent retraction and recurrent cholesteatoma, multivariate analysis shows that the other risk factors have a more significant effect on the development of recurrence. These findings should be considered when choosing the surgical approach for paediatric cholesteatoma.

## Figures and Tables

**Figure 1 jcm-13-00836-f001:**
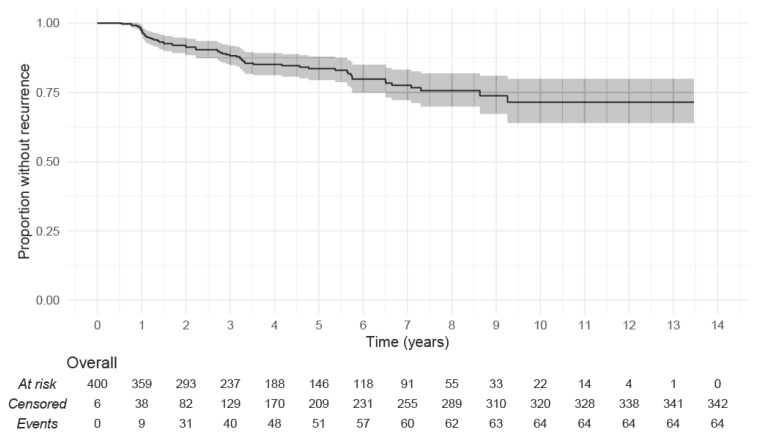
Kaplan–Meier plot showing the proportion of all ears surviving without recurrence of cholesteatoma over time. 95% confidence intervals in grey shadow.

**Figure 2 jcm-13-00836-f002:**
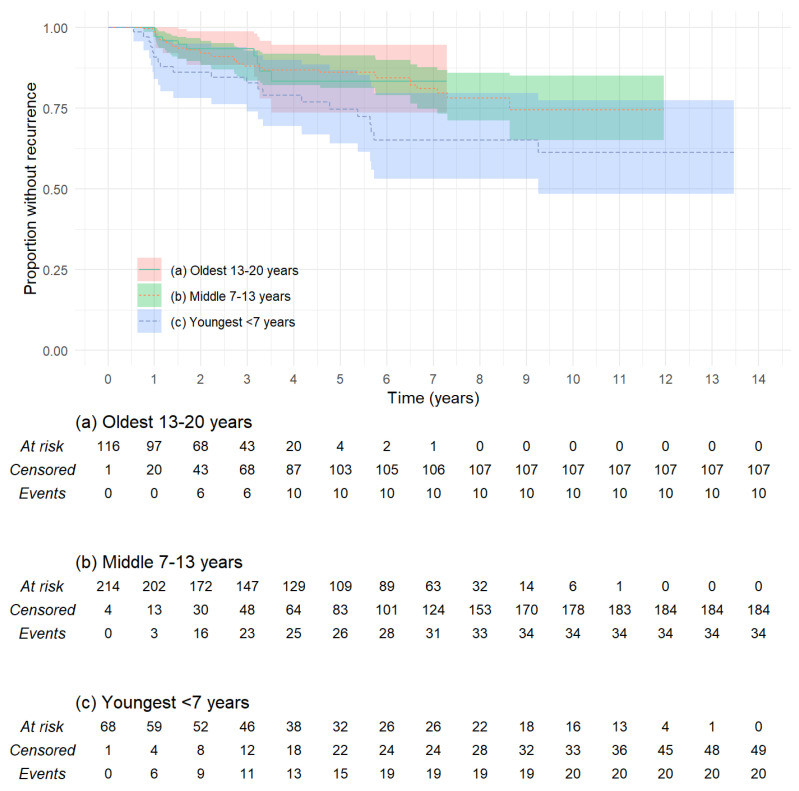
Kaplan–Meier plot comparing rate of recurrence in different age groups, grouped as oldest (age 13–20 years), middle (7 to <13 years), and youngest (under 7 years). The youngest children have greater risk than older children (log-rank *p* = 0.075).

**Figure 3 jcm-13-00836-f003:**
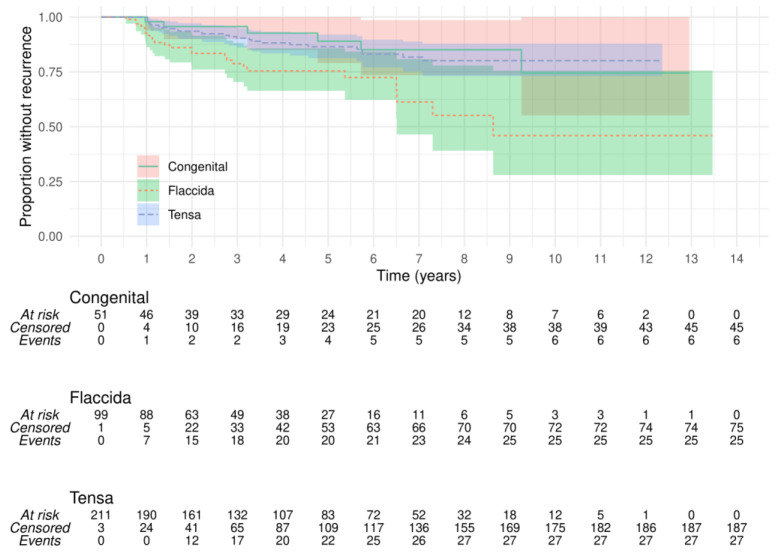
Kaplan–Meier plot showing the time course of development of recurrent cholesteatoma after removal of congenital, pars flaccida, or pars tensa cholesteatoma. The risk was significantly greater for recurrence of pars flaccida cholesteatoma with this univariate analysis (log-rank, *p* = 0.001) compared with tensa or congenital which were not significantly different.

**Figure 4 jcm-13-00836-f004:**
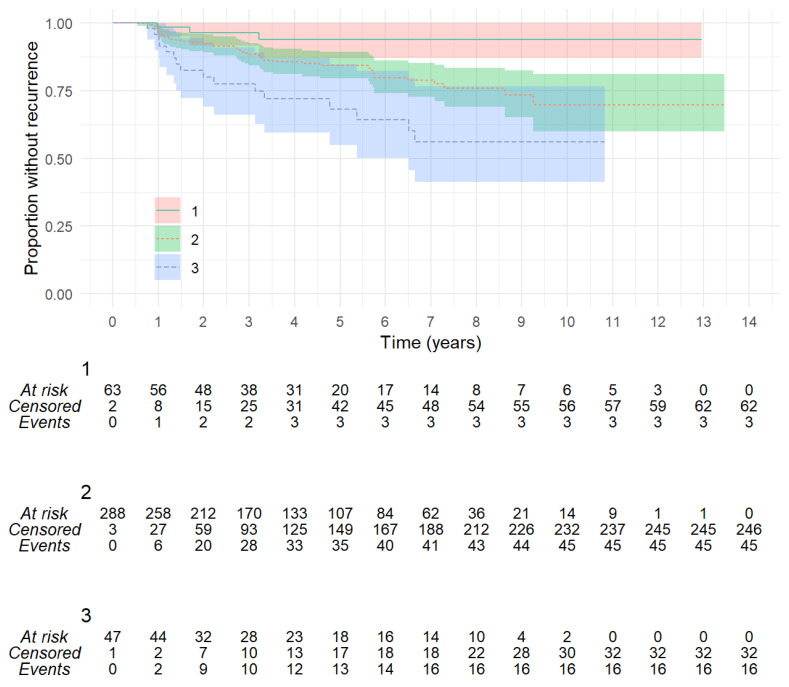
Kaplan–Meier plot of recurrent cholesteatoma according to EAONO-JOS stage. A clear increase in risk with increasing stage is visible. Stage 1, disease limited to the subsite of origin, has least risk. Stage 2 includes cholesteatoma in two or more subsites without other markers of severity. Stage 3 includes cholesteatoma causing more erosive disease and has the highest risk (log-rank, *p* = 0.0011).

**Figure 5 jcm-13-00836-f005:**
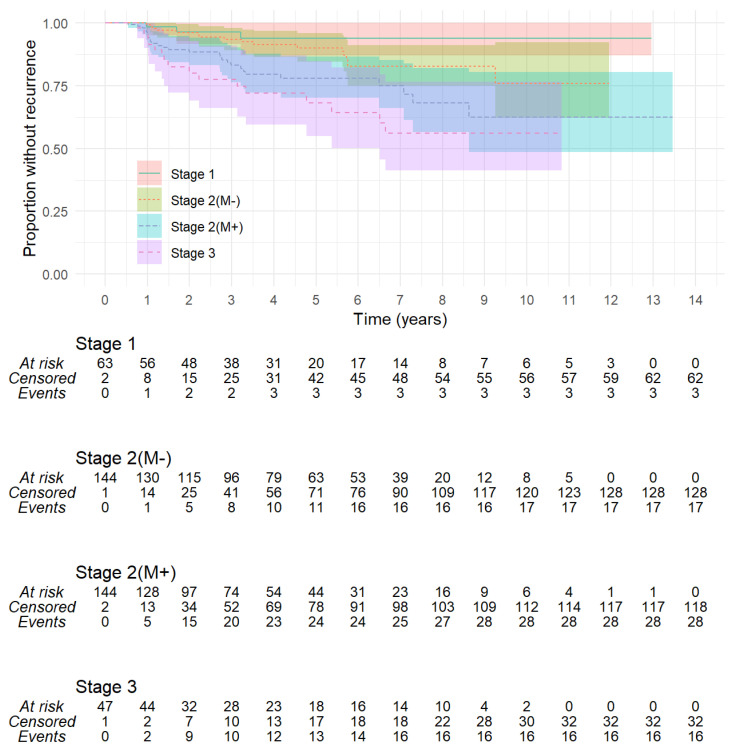
In this Kaplan–Meier plot, EAONO-JOS Stage 2 has been subdivided according to absence (M−) or presence (M+) of cholesteatoma in the mastoid. Increasing risk of recurrence is visible for larger and more erosive cholesteatoma (log-rank, *p* = 0.00027).

**Table 1 jcm-13-00836-t001:** Multivariate analysis of variables on rate of recurrent cholesteatoma (Cox proportional hazard analysis). Based on findings from univariate Kaplan–Meier analysis, selected comparisons are youngest age group (under 7 years age) compared against older children; children with syndromic risk factors against those without; pars flaccida against other cholesteatoma types; EAONO-JOS Stage 2 without (M−) or with (M+) mastoid cholesteatoma and Stage 3 against Stage 1; surgical category Group b (canal wall-up tympanomastoidectomy) and Group c (mastoid obliteration and/or canal wall-down surgery) against Group a (trans-canal mastoid-sparing surgery). * significant variables.

Variable	Hazard Ratio	95% Confidence Interval	*p*-Value
Youngest age (<7 years)	1.90	1.09–3.30	0.023 *
Risky Syndrome	2.88	1.10–7.50	0.030 *
Pars flaccida	0.50	0.97–2.81	0.054
EAONO-JOS Stage 2 (M−)	2.14	0.62–7.43	0.231
EAONO-JOS Stage 2 (M+)	3.20	0.86–11.89	0.083
EAONO-JOS Stage 3	5.06	1.38–18.53	0.014 *
Surgical category b	1.66	0.88–3.12	0.114
Surgical category c	0.90	0.35–2.31	0.827

## Data Availability

The datasets presented in this article are not readily available because of privacy constraints. Requests to access the datasets should be directed to the author.

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
