# Peer review of "Cholesteatoma Severity Determines the Risk of Recurrent Paediatric Cholesteatoma More Than the Surgical Approach"

_jcm, 2024, doi:10.3390/jcm13030836_

Round 1

Reviewer 1 Report

Comments and Suggestions for Authors

lines 80, 90, 97 abbreviations are not explained

usually in congenital cholesteatomas there are residuals and not recurrence. the same probability of recurrent disease that the authors observed in congenital and tensa acquired cholesteatoma should be explained

A well done canal wall down tympanomastoidectomy doesn't leave space for recurrent colesteatoma, whereas canal wall up with mastoid obliteration is still at risk for recurrent disease, so I think that they shouldn't be grouped together. Canal wall down will show significant less recurrent disease.

Author Response

Thank you for completing the review and for your comments.

1) Explanation of the abbreviations has been added. The SAMEO-ATO nomenclature is now an internationally accepted standard used to summarise the type of surgery more clearly than traditional nomenclature. The original description is referenced for readers that require more information.

2) I appreciate the reviewer's appropriate distinction between residual and recurrent disease which is often confused in published literature. However, as summarised in the discussion, there is a common misconception that retraction disease should not be expected after removal of congenital cholesteatoma. We and others have found consistently that this does occur, as referenced and explained in the discussion: "It is of note that acquired retraction cholesteatoma can develop after removal of congenital cholesteatoma. Although not anticipated from current understanding of cholesteatoma pathogenesis, this finding has been reported previously from different centres and shows that follow up of congenital cholesteatoma should be maintained for several years to monitor for this risk of retraction sequelae [20,21]." I cannot explain why this happens, but this is what the data show.

3) A well done canal wall down tympanomastoidectomy leaves space in the protympanuma and sinus tympani for recurrent colesteatoma. The only extra space present after a CWU obliteration is the facial recess. In my opinion this would not be expected to create a significantly increased risk of recurrence. Advocates of CWU obliteration claim very low rates of recurrence (eg van Dinther et al 2015, Hellingman et al 2019), even lower than after CWD surgery (eg van der Toom et al 2021). My own data (including revision of surgeries completed elsewhere so excluded from this analysis) show similar recurrence rates after CWD surgery. So I respectfully disagree with the reviewer. The functional similarity of the ventilated middle ear with attic and mastoid removed from the ventilate space make CWD and obliterated CWU sufficiently similar to justify amalgamation for this analysis.

Reviewer 2 Report

Comments and Suggestions for Authors

Overall, this is a well-written review in which children from a single surgeon who performed cholesteatoma resection were analyzed for risks of recurrence. This manuscript found that age and severity of cholesteatoma at the time of surgical presentation are the most important risk factors. Type of surgery performed is less relavent. Although many of these findings are self-evident, the surgical approach not being a risk factor is of interest. This is relevant because transcanal endoscopic surgery is becoming more and more popular, and if this surgery was found to be a risk factor for recurrence, this would be of great importance. This was not the case. 

The main weakness of the paper, that it is of a single surgeons experience, is also a strength. The single surgeon implies uniformity in surgical decision-making. Although this may limit the patient population that the patient sees, it may control for other external factors such as surgical judgement in choosing approach, etc. 

Overall, this is a well written article that adds to the literature. 

Minor edits: 

Line 287 should say 'recurrence' 

Author Response

Thank you very much for this positive review.

Also for spotting the typo on line 287 which has been corrected

Reviewer 3 Report

Comments and Suggestions for Authors

This is a well written and designed study with impressive numbers and follow up.   As pointed out by yourselves, these results have some surprising findings, mostly that a more aggressive approach , that is the canal wall down procedure did not have any better results than the more conservative approaches but the number in this category was also much smaller that could have impacted on the ultimate results.

Author Response

Thank you very much for this positive review

Round 2

Reviewer 1 Report

Comments and Suggestions for Authors

If a retraction pocket develops after removal of most congenital cholesteatomas (particularly if middle ear is the only localization), it should be considered a primary cholesteatoma, not a recurrent cholesteatoma.

Middle ear space after CWU obliteration is larger than after CWD, so the only explanation of recurrent cholesteatoma is that CWD was not properly performed.

Author Response

Thank you for these comments.

1)  The use of nomenclature for retraction cholesteatoma after congenital cholesteatoma is a matter for semantic debate. As most people don't realise that acquired retraction cholesteatoma can occur after removal of congenital cholesteatoma, there is no accepted nomenclature for this condition. By one definition, cholesteatoma that develops from retraction after previous complete removal is a recurrence (regardless of the initial type of cholesteatoma). By another definition, development of an acquired retraction after a previous congenital anomaly is a new pathological process so is not a true recurrence of the intial problem. 

The definition of recurrence used in the manuscript is clarified in the methods section:

"Recurrent cholesteatoma was defined as an accumulation of keratin debris from squamous epithelium that was in continuity with the surface of the tympanic membrane and could not be removed with microdebridement. Usually this was from a new tympanic membrane retraction pocket. Growth of skin into previously obliterated bone and entrapment of skin from new bone growth across a mastoid cavity were included within this definition. An acquired cholesteatoma from retraction after removal of a congenital cholesteatoma was also classified as recurrence."

In recognition of the contradiction in the two different semantic definitions, mention of this condition has been changed in the results and discussion from recurrence after congenital cholesteatoma to "development of retraction cholesteatoma after congenital cholesteatoma".

2) May be the CWD surgeries were not completed properly, but the recurrence still occurred. There are additional explanations. The definition of recurrence used in this manuscript was defined in the methods section, as above. 

One of the additional problems found in paediatric CWD that perhaps does not happen so often in adults, is that of new bone growth from the mastoid cortex, facial ridge of tegmen that closes off access to the mastoid cavity. Squamous epithelium is then trapped in an area that cannot be cleaned (aka cholesteatoma). Again we are limited by available definitions of recurrence: this is disease that is in continuity with the surface of the remanant tympanic membrane so by definition is a recurrence. It is not a remanant of the initial disease (so is clearly not residual). However it has not been formed by negative middle ear pressure and retraction, the mechanism more commonly assumed to cause recurrence. 

Similarly, some of the CWD surgeries developed "recurrence" into the mastoid antrum which have previoulsy been obliterated. That is, skin grew into an area that had been covered with soft tissue and filled with bone pate. Again it could be argued that the obliteration was not done properly, but nevertheless the process occured. It is a regrowth of a new cholesteatoma that is not residual disease, so best fits into the recurrence category.  This process can also occur after CWU obliteration. 

The reviewer may feel that the term "recurrence" should only be applied to retraction pockets into a ventilated space caused by negative pressure. However I do not feel this restricted use of the term is appropriate as (1) there is no mechanism to determine the cause of the pathophysiological process; (2) there is no alternative term to refer to the presumed different mechanisms of recurrence; 3) internationally accepted definition is "Recurrent cholesteatoma results from the reformation of the retraction pocket after a complete previous surgical cholesteatoma removal." It does not state that the retraction pocket must grow into a ventilated space. 

Perhaps revised definitions of recurrence are required to distinguish between the different processes highlighted by the reviewers comments, that might justify an accompanying editorial, but that is beyond the scope of this manuscript.

In response to the academic editor, additional text has been added to the text highlighting the similarities of cases in which the mastoid has been separated from the ventilated middle ear space.

""This group includes ears in which the mastoid air space is disconnected from the middle ear air space. ... Both canal wall down and canal wall up obliteration eliminate the mastoid air space and create a small middle ear space, in principle leaving similar spaces into which recurrent disease could grow. With all of the techniques in this group, gas absorption across mastoid mucosa and any pressure buffering effect of the mastoid are removed so cannot contribute to middle ear pressure homeostasis and retraction disease. Predisposition to recurrent cholesteatoma is therefore considered equivalent."

While it is true that the middle ear space is slightly smaller after CWD than CWU obliteration as the facial recess is removed in CWD. However the protympanic, hypotympanic and medial retrotympanic spaces are identical. 

Ultimately, the finding that recurrence rates are similar between these different surgeries (based on a larger dataset including cholesteatoma receiving initial surgery elsewhere) shows that there is sufficient homogeneity in outcome to justify grouping them together, regardless of theoretical differences.